# Trend of Antimicrobial Use in Food-Producing Animals from 2018 to 2020 in Nepal

**DOI:** 10.3390/ani13081377

**Published:** 2023-04-17

**Authors:** Nabin Upadhyaya, Surendra Karki, Sujan Rana, Ibrahim Elsohaby, Ramanandan Tiwari, Manoj Oli, Surya Paudel

**Affiliations:** 1Veterinary Standards and Drug Regulatory Laboratory, Budhanilakantha, Kathmandu 44600, Nepal; drnabinvet@gmail.com; 2Food and Agriculture Organization of the United Nations, Emergency Center for Transboundary Animal Diseases, Lalitpur 44700, Nepal; karkisuren@gmail.com; 3Department of Livestock Services, Hariharbhawan, Lalitpur 44700, Nepal; sujanrana@gmail.com; 4Department of Infectious Diseases and Public Health, Jockey Club College of Veterinary Medicine and Life Sciences, City University of Hong Kong, Hong Kong SAR, China; ielsohab@cityu.edu.hk; 5Centre for Applied One Health Research and Policy Advice (OHRP), City University of Hong Kong, Hong Kong SAR, China; 6Ministry of Agriculture and Livestock Development Nepal, Singhadurbar, Kathmandu 44600, Nepal; tiwarinlbc@gmail.com; 7Tulsipur Municipality, Tulsipur, Dang 22412, Nepal; vetmanoj2016@gmail.com

**Keywords:** antimicrobial use, Nepal, antibiotics, veterinary, food-producing animals

## Abstract

**Simple Summary:**

The inability of antimicrobials to kill or inhibit the growth of microorganisms that they used to previously kill or inhibit leads to antimicrobial resistance (AMR). Overuse of antimicrobials is usually associated with AMR; thus, global efforts have been made to systematically monitor the availability and use of antimicrobials in humans and animals. In this study, a survey was conducted from 2018 to 2020 to estimate the actual availability of different antimicrobials for veterinary use in Nepal, targeting major government and non-government stakeholders relevant to the authorization, production, and sales of antimicrobials. It was found that the total availability of antimicrobials, including class I antibiotics that are critically important for the treatment of human diseases, remarkably declined with the progression of time, meaning less antibiotics were used for the treatment of animals in 2020 than in 2018 in Nepal. As animal health is closely connected with human health, the data are very encouraging from a public health point of view. However, the awareness and organized surveillance should be continued in the future.

**Abstract:**

Antimicrobial resistance is a global public health problem and is primarily driven by the widespread overuse of antibiotics. However, antimicrobial use data in animals are not readily available due to the absence of a national database in many developing countries, including Nepal. This study was conducted to estimate the quantities of antimicrobials available in Nepal as an indicator of their use in food-producing animals between 2018 and 2020. Data were collected through surveys targeting major stakeholders: (i) the Department of Drug Administration (DDA), the Government of Nepal (GoN) for the authorized antimicrobials for veterinary use in Nepal, (ii) veterinary pharmaceuticals for antimicrobials produced in Nepal, (iii) the DDA and Veterinary Importers Association for antimicrobials bought by veterinary drug importers, and (iv) the Department of Customs, GoN, for antibiotics sourced through customs. Data showed that in the 3 years, a total of 96 trade names, comprising 35 genera of antibiotics representing 10 classes, were either produced or imported in Nepal. In total, 91,088 kg, 47,694 kg, and 45,671 kg of active ingredients of antimicrobials were available in 2018, 2019, and 2020, respectively. None of the antibiotics were intended for growth promotion, but were primarily for therapeutic purposes. Oxytetracycline, tilmicosin, and sulfadimidine were among the most-used antibiotics in Nepal in 2020. Oxytetracycline was primarily intended for parenteral application, whereas tilmicosin was solely for oral use. Sulfadimidine was available for oral use, except for a small proportion for injection purposes. Aminoglycosides, fluroquinolones, nitrofurans, sulfonamides, and tetracyclines were mostly produced locally, whereas cephalosporins, macrolides and “other” classes of antimicrobials were imported. Amphenicols and penicillins were exclusively imported and nitrofurans were produced locally only. In general, except for tetracyclines, the volume of antimicrobials produced locally and/or imported in 2020 was lower than that in 2018, which corresponded to a decreasing trend in total antimicrobials available. Furthermore, the subsequent years have seen a decrease in the use of critically important antibiotics, particularly class I antibiotics. Finally, this study has firstly established a benchmark for future monitoring of antimicrobial usage in food-producing animals in Nepal. These data are useful for risk analysis, planning, interpreting resistance surveillance data, and evaluating the effectiveness of prudent use, efforts, and mitigation strategies.

## 1. Introduction

The livestock sector is intricately connected with the social, economic, nutritional, and cultural dimensions of Nepal, and diverse animal species such as cattle, buffaloes, sheep, goat, pigs, and poultry are raised at a commercial or semi-commercial scale, which altogether significantly contributes to the household economy and food security [1]. The average supply of protein of animal origin in Nepal has increased sharply over the last two decades, and the trend is expected to rise further due to the increase in human population and per capita income [2]. However, the growing livestock sector is also challenged by the high prevalence of infectious diseases, both zoonotic and non-zoonotic, limiting productivity, and thus increasing the vulnerability of the livestock industry [3]. Mitigation of many such diseases might require the use of antimicrobials, which are a known contributor of antimicrobial resistance (AMR). AMR is commonly defined as the inability of antimicrobials to inhibit the growth of bacteria that they used to inhibit previously, and is regarded as one of the global health crises of the present era [4,5]. A growing livestock industry driven by the continuous demand for animal protein as a widely affordable source of food might promote the use of antimicrobials for the treatment of livestock diseases, growth promotion, or as low-cost substitutes for hygiene measures. Furthermore, farmers and prescribers are often found using antimicrobials even when they are not actually required due to a lack of awareness and poor regulatory measures. Collectively, the higher use of antimicrobials is a confounding factor contributing towards AMR [6]. Furthermore, increasing AMR in zoonotic pathogens that may not necessarily cause diseases in animals, such as *Salmonella* and *Campylobacter* [7], and the dissemination of such resistant bacteria and/or AMR genes to humans [8], are even more concerning. If no actions are taken, the global sales and use of antimicrobials are expected to rise until 2030 on all continents, primarily in developing countries, as commercial livestock growth is higher in these countries [9]. Thus, judicial use of antimicrobials in animals undertaking the concept of One Health, which recognizes the interdependency among human, animal, and environmental health is very crucial to combat the situation and minimize the food safety risks [10].

Regular surveillance and monitoring of antimicrobial use (AMU) are essentially required to fight against AMR. The measurement of AMU in human and animal health production settings is one of the major goals of the Global Action Plan on Antimicrobial Resistance of the World Health Organization (WHO) [11], as well as the complementary plans and strategies developed by the Food and Agriculture Organization of the United Nations (FAO) and the World Organization for Animal Health (WOAH, formerly known as OIE) [12]. Based on this strategy, the WOAH, supported by the FAO and WHO within the tripartite collaboration, has taken a lead to build a global database on antimicrobial agents intended for use in animals. Nonetheless, there are considerable data gaps regarding AMU in different animal species and humans in most countries. This knowledge gap is due to the absence of reliable AMU data in humans and animals and ill-defined animal population denominators. Many high-income countries, such as member states of the European Union (EU), regularly publish their data on AMU in different sectors, and relate these values to denominator populations in terms of biomass [7]. The European Surveillance of Veterinary Antimicrobial Consumption (ESVAC) was launched in 2009 and has been reporting data from 31 member states of the European Union [13]. Conversely, no such strong mechanism exists to regularly collect and report equivalent AMU statistics in many other nations, especially in low- and middle-income countries (LMICs). Outside the EU, only nine countries regularly publish national reports on AMU. The WOAH has been collecting voluntary data on the use of antimicrobial drugs in animals annually since 2016, and the participating nations have reached up to 157 [14,15]. However, the list of countries that provide data on AMU for each region is not made public. As a result of the unidentifiability of the countries that submit data to the WOAH and the regional aggregate of national AMU reports, it is difficult to monitor antimicrobial use at a national level.

So far, as in many developing countries, Nepal lacks its own formal national database system on antimicrobial use in livestock. Lack of such a mechanism at a national level might influence the government policies and stewardship initiatives to curb AMU. It may also hinder additional epidemiological studies related to the AMU on a national scale, such as determining a country’s antibiotic footprint using national trade data [16]. On a positive note, in Nepal, despite a lack of a formal national database on AMU in animals, Veterinary Standards and Drug Regulatory Laboratory (VSDRL), a focal office responsible for AMU reporting to WOAH, has been collecting data on AMU through its network to report to the WOAH annually.

The objectives of this study were (i) to share a mechanism of data collection involving different stakeholders to reliably estimate the quantities of antimicrobials available for use in food-producing animals in Nepal, and (ii) to understand the temporal trend of AMU in food-producing animals in Nepal between 2018 and 2020. For this, the study included various aspects of antimicrobial consumption, such as antimicrobials authorized for use in Nepal, as well as the volume of antimicrobials imported, produced, and sold for veterinary use. The study is the first of its kind in Nepal to show the trend in antibiotic availability at the national level for use in the livestock sector based on reliable data sources, which can be used as a benchmark for future monitoring of antimicrobial use in Nepal and beyond.

## 2. Materials and Methods

### 2.1. Data Sources

Data were collected from organized surveys conducted at various levels in the years 2018, 2019, and 2020 in Nepal. The surveys comprised three primary sections, namely: (i) demographic information; (ii) general information about AMU in animals for both growth promotion and therapeutic purposes; and (iii) data collection of the antimicrobial agents intended for use in animals. Within this section, the data sources, animal groups, and route of administration were collected. Data from different sources were carefully analyzed to avoid double counting. The data utilized in this study were obtained from various sources intended for different purposes, as follows:(a)Range of antimicrobials authorized for veterinary use in Nepal

This information was obtained from the Department of Drug Administration (DDA), Government of Nepal, which is the national competent government authority for registration of all antimicrobials for use in humans and animals under the Drug Act (1978) [17].

(b)Antimicrobial volumes sourced from pharmaceutical companies

Information on the quantity of antimicrobials manufactured and supplied by the veterinary pharmaceutical industry in Nepal for veterinary use were sourced from all eight veterinary pharmaceutical companies registered in the DDA.

(c)Antimicrobial volumes sourced from veterinary drug importers

Information on the quantity of antimicrobials imported by veterinary importers in Nepal for veterinary use were obtained from the DDA and Veterinary Importers Association, Nepal. For this, a survey template was developed based on the guidelines of the WOAH for reporting AMU, and the form was distributed to the importers. The collected data were analyzed later, as explained below.

(d)Antimicrobial volumes sourced from customs

Information on the quantities of antimicrobials imported for veterinary use by the veterinary importers registered in the DDA, Nepal was obtained from the Department of Customs under the Government of Nepal.

### 2.2. Data Collection and Observations

Quantities (number of vials or packs) including potency of medicinal preparations imported for sale or manufactured for local use were recorded for each antimicrobial class or combination product. Similar was performed for the data obtained from the customs for each product. The observations for the authorized antibiotics included the categories such as name of importing or manufacturing company, date of import or manufacture, brand or trade name of the product, nonproprietary name (NPN) or generic name, dosage form or route of administration (parenteral/oral route), and quantity of active ingredient of each antibiotics class imported or manufactured per product. Antibiotics were then classified as aminoglycosides, amphenicols, cephalosporins (all generations), fluoroquinolones, macrolides, nitrofurans, penicillin, sulfonamides (including trimethoprim), tetracyclines, and others (e.g., tylosin, tiamulin, bacitracin, and tilmicocin) as per the WOAH Annual Report [14]. The quantities of all antimicrobial ingredients in the drug preparations were then calculated and entries were prepared in the data sheet.

### 2.3. Trustworthiness of the Source of Data

All information collected from different sources can be reasonably regarded as trustworthy, as they were obtained from the official government sources and were collected for official reporting to the WOAH. In this study, data were collected mainly from importers and manufacturers, who represent the grassroots level or the tier I distribution system. Although it can be argued that data from such sources may not be accurate for intended target animal and route of administration, in this pharmaco-epidemiological study, this method or data collection is probably the best in the local context because retailers, veterinarians, and producers do not keep records of end use data due to inadequate enforcement by regulatory authorities. The data were collected by a competent authority (WOAH National Focal Point for Veterinary Products) through email, and the required information were clearly explained to the data providers. The information was checked thoroughly, and, if necessary, required amendments were made after discussion with the respective data providers.

### 2.4. Reliability of the Data

The reliability of the data was confirmed by obtaining information from the Department of Customs, which supplied the same template of data from which the same analyses and deductions could be made, as undertaken from other sources.

### 2.5. External Validity of the Data

The results of this study are transferable to any other data analyst because of the organized data collection procedure.

### 2.6. Internal Validity of the Data

The study was undertaken in accordance with the approved protocol and the measurements were consistently taken in the same way from the information supplied. The only information that could not be measured was the information on the availability of antimicrobials sourced from illegal trade, which cannot be overruled due to the ban of antimicrobials as growth promoters and the porous border of the country.

### 2.7. Data Analysis

All the collected data were entered in the Microsoft Excel program and quantities for specific variables such as antimicrobial classes, types, amount, target animals, and intended usage were calculated. Amount of antimicrobial agents were calculated in kilogram active ingredient for each antimicrobial class. Data visualization was carried out using R software (R Core Team, 2022; version 4.2.0).

## 3. Results

### 3.1. Availability of Authorized Antimicrobials for Veterinary Use in Nepal

The total volume of antimicrobials available for use in food-producing animals in Nepal is determined by the sum of amount imported and internally produced by the local pharmaceutical companies in the country. This matric serves as a direct reflection of the pattern of AMU. The study identified a total of 96 trade name registered antimicrobials available for use in animals, which comprised 35 different genera of antibiotics and belonged to 10 different classes. “Others” include tylosin, tiamulin, bacitracin, and tilmicosin. All these antimicrobials were reported to be used for treatment and prophylactic purpose, but not for growth promotion. The numbers of antimicrobial classes and corresponding amount available in Nepal in the years 2018, 2019, and 2020 for veterinary use are shown in Table 1. Overall, there was a gradual decline in the availability of veterinary antibiotics in Nepal over the three-year time period, importantly class I antibiotics, remarkably cephalosporins, aminoglycosides, and fluroquinolones (Table 1). In 2018 and 2019, 91,088 kg and 47,694 kg of active ingredients of antibiotics were available, respectively. In 2020, the availability further declined to 45,671 kg of active ingredients, which is half of the amount reported in 2018. Amphenicols were found to be the least available antibiotics class in all three years.

In 2020, oxytetracycline, tilmicosin, and sulfadimidine were ranked among the top antimicrobial agents reported (Figure 1), which together accounted for 54.4% of the total availability. Oral use of antimicrobials shared a large proportion of the total antimicrobials available (Figure 2). Among the top three antimicrobial agents available, oxytetracycline was majorly intended for parenteral application, whereas tilmicosin was solely for oral use. Sulfadimidine was available for oral use, except for a small proportion for injection purposes.

### 3.2. Antimicrobials Imported and Internally Produced for Veterinary Use in Nepal

Among antimicrobials produced by the local pharmaceutical companies, aminoglycosides, fluroquinolones, nitrofurans, sulfonamides, and tetracyclines shared a substantial proportion, whereas cephalosporins, macrolides, and “other” classes of antimicrobials were minimally produced as compared with the imported volume (Figure 3). Amphenicols and penicillins were exclusively imported and nitrofurans were only produced locally. In general, the volume of antimicrobials produced locally and/or imported was lower in 2020 compared with that in 2018, with the exception of tetracyclines, which corresponded with a decreasing trend of total antimicrobials available, as shown in Table 1.

## 4. Discussion

The overall demand for animal-source nutrition through 2030 is anticipated to be higher in LMICs as compared with high-income countries. For instance, the projection of poultry farming between the year 2000 and 2030 is expected to rise by 725% [19]. Thus, livestock is becoming one of the fastest growing agriculture sectors in South Asia. Consequently, antimicrobial consumption in animal production systems in LMICs is growing, as well as commensuration with expanded intensive production to meet rapidly increasing demand for animal-source nutrition, but AMU remains largely undocumented [20]. Furthermore, as an example in poultry production, the appearance of diseases in the field is multifactorial, influenced by the coexistence of predisposing factors, among others [21,22,23]. Within the same bacterial species isolated from the same bird, very high heterogenicity in terms of antimicrobial sensitivity profiles can be observed in different colonies [24,25]. From a clinical point of view, such factors might pose additional challenges for successful treatment of bacterial diseases with antibiotics, which could ultimately trigger higher AMU. In addition, several antimicrobials are used in livestock for non-therapeutic purposes, such as growth promotion and prevention of diseases. The use of antimicrobials is a confounder of AMR, both in humans and livestock. Finally, the antimicrobial resistant microbes, either pathogenic or commensal, can transfer to humans through contaminated food (farm-to-fork transmission) [26]. In LMICs, the risk of transmission of antimicrobial resistance from animals to humans is higher due to unique livestock practices, where humans are in greater contact with animals and their natural environment, poor biosecurity measures in farms, informal livestock trade, and insufficient veterinary services [27]. Thus, from a public health perspective, it is extremely important to understand the situation of AMU at the source, such as livestock production. Under antibiotic stewardship, the Government of Nepal started reporting the availability of antimicrobials in veterinary medicine to the WOAH, using a specified format provided. The data collected in this study were intended for the same purpose. Starting from 2020, the report format has been changed, which now includes more detailed information than in the previous two years, such as specific antimicrobial agents, routes of application, etc.

Acknowledging the importance of proper documentation of AMU, several countries around the globe have adopted a defined channel for the sale and use of antimicrobials. As an example, in the EU, all the veterinary antimicrobial agents must be sold through distributors authorized by the competent authority in each country. In Australia, the Therapeutic Goods Administration (TGA) records all the types and volume of antimicrobials for human and veterinary use. The Austrian Poultry Health Service (QGV) has established a Poultry Health Database (PHD) in which each and every antimicrobial treatment in poultry is recorded, allowing deeper insights of AMU in individual flock [28]. With such strict regulations, it is easily possible to identify all distributers of antimicrobials and gather 100% of the data on AMU and sales. In developing countries, the data on the actual use of antibiotics in animals in the field are often difficult to retrieve due to a lack of proper functioning drug regulatory mechanisms by the end users [29]. In Nepal, there are several stakeholders such as producers, importers, distributers, prescribers, and end users who all are associated with the proper use of antibiotics. Thus, in this study, multiple sources were considered to collect data, which can be reasonably suggested as an exemplary model for future surveillance programs in developing countries to collect valid information on AMU.

The present study showed that all the main classes and types of antibiotics were found to be authorized for food animal use in Nepal under the drug act. As Nepal has a legal ban on the use of antimicrobials as growth promoters, all antibiotics were intended for treatment purposes. Furthermore, the data showed that the stakeholders related to livestock production in Nepal have made substantial progress towards the goal of reducing overall use of antibiotics in animals. The progress should have been possible due to government policies to curb AMU, as well as increasing awareness of AMR among farmers, veterinarians, and consumers. The use of critically important class I antibiotics in livestock decreased substantially, which is a very positive sign. However, not to forget, with regard to antimicrobial use, less can be more. A cross sectional study showed that the bacterial pathogens in humans and animals in Nepal are becoming highly resistant to the first- and second-line antibiotics [30]. In a recent study from Nepal, the presence of resistant genes against several antimicrobial classes such as aminoglycosides, beta-lactams, tetracyclines, macrolides, and fluoroquinolones were found in samples from humans, animals, and the environment [31]. Thus, efforts should be further continued to decrease the AMU to the minimum possible level. In this regard, focus should also be on enforcing good husbandry and management practices, as well as enhancing internal and external biosecurity in livestock farms that can significantly lower the AMU [32].

So far, there are no legislative mandatory requirements for pharmaceuticals and importers to report antimicrobials produced/imported in Nepal, and such information is considered very sensitive. Thus, data in this study were obtained with great difficulties through official request and personal communication. However, it might be worth reviewing the existing legislation that regulates veterinary products and include mandatory mechanisms to report annual antimicrobials produced and imported, along with the actual use in the field, from relevant stakeholders. Considering the importance of antimicrobial regulation, stringent control channels could be implemented involving customs and Nepal Rastra Bank, which is the supreme banking institution and a watchdog for foreign investment and trade in Nepal. The tariff codes established by customs would have to be revisited and made much more specific for the different classes of veterinary antimicrobials. These AMU estimates could be fine-tuned by conducting targeted surveys tailored to different production types (i.e., meat chickens, layers, breeders, fattening pigs, etc.). It may also be necessary to differentiate the extent of AMU by the level of intensification of the production system (i.e., backyard, small-scale, large-scale, or industrial), as different systems require variable quantities of antimicrobials. Furthermore, it is also very important that such surveillance systems regarding the consumption of antimicrobials be paired with the prevailing antimicrobial resistance pattern, and, if necessary, amend the regulations for specific drugs. Currently, several governmental and non-governmental institutions such as the National Public Health Laboratory and the Department of Livestock Services have been conducting antimicrobial resistance surveillance in Nepal.

One of the limitations of this study is that the approach of data collection does not allow us to understand the exact exposure of individual animal populations to specific antimicrobials. Furthermore, this study does not show the spatial pattern of antimicrobial use in different areas of Nepal. Since it is not mandatory for farmers/end users in Nepal to record the antimicrobials used in their animals, which might have been treated with different antimicrobials and doses for different durations, obtaining such data is rather difficult until strict regulations are implemented by the government. Secondly, it was only intended to obtain data regarding antimicrobial consumption. Antimicrobial residues in animal products and antimicrobial resistance patterns in bacteria isolated from different sources were not tested, which were not the scope of this study. Thus, the direct correlation between the use of antimicrobials and actual resistance pattern seen in the field in the particular time period cannot be made, but is suggested for future consideration. Despite the above-mentioned limitations, this study provides valid and robust information regarding AMU in Nepal. Due to the difficulties experienced during data collection, it is also suggested that a software database should be developed to facilitate a harmonized monitoring system of AMU in Nepal in the future.

## 5. Conclusions

This study showed that the overall consumption of antimicrobials, including the critically important class I antibiotics, in food-producing animals in Nepal declined from 2018 to 2020. It has established a foundation for future monitoring systems to understand antimicrobial usage in food-producing animals and creating a national database on the use of antimicrobials in livestock sector in Nepal. These data are useful for risk analysis, planning, interpreting resistance surveillance data, and evaluating the effectiveness of prudent use efforts and mitigation strategies.

## Figures and Tables

**Figure 1 animals-13-01377-f001:**
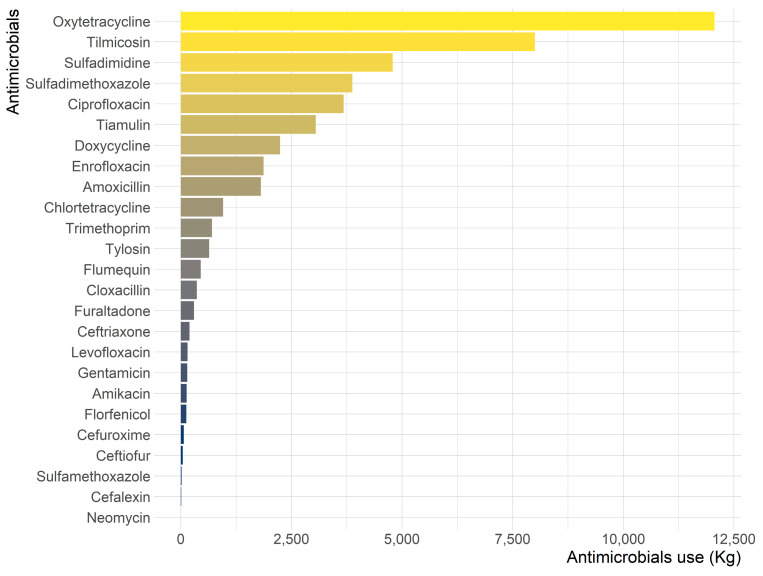
Antimicrobials quantities (by antimicrobial agents) reported for use in veterinary medicine in Nepal in 2020.

**Figure 2 animals-13-01377-f002:**
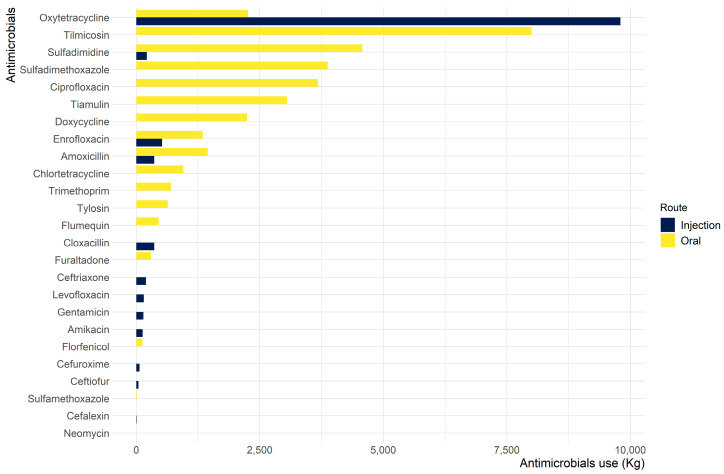
Antimicrobial quantities (by antimicrobial agents and route of administration) reported for use in veterinary medicine in Nepal in 2020.

**Figure 3 animals-13-01377-f003:**
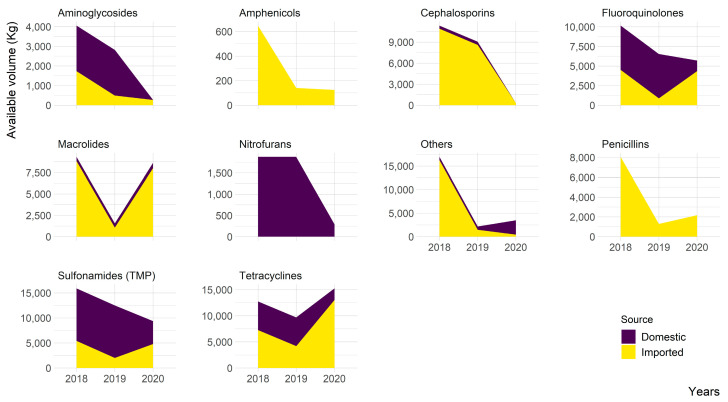
The volume of imported and domestic antimicrobials available in Nepal for veterinary medical use between 2018 and 2020 (Others include tylosin, tiamulin, bacitracin, and tilmicocin).

**Table 1 animals-13-01377-t001:** Veterinary medical use of antimicrobials in Nepal between 2018 and 2020.

Rank ^1^	Antimicrobial Class	Antimicrobial Use in Kg *		
		2018	2019	2020
I	Cephalosporins	11,355	9061	319
I	Aminoglycosides	4056	2820	275
I	Amphenicols	648	141	125
I	Fluoroquinolones	10,162	6535	5702
I	Macrolides	9329	1575	8639
II	Penicillins	8051	1296	2174
II	Sulfonamides (including TMP) ^2^	15,916	12,512	9382
II	Tetracyclines	12,769	9696	15,257
III	Nitrofurans	1875	1875	297
	Others ^3^	16,927	2184	3501
Total		91,088	47,694	45,671

* Active ingredients; ^1^ rank I, critically important; rank II, highly important; rank III, important (based on World Health Organization’s categorization) [18]; ^2^ TMP: trimethoprim; ^3^ Others includes tylosin, tiamulin, bacitracin, and tilmicocin.

## Data Availability

The data used in the present study are available in the manuscript.

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
