# Peer review of "Trend of Antimicrobial Use in Food-Producing Animals from 2018 to 2020 in Nepal"

_animals, 2023, doi:10.3390/ani13081377_

Round 1

Reviewer 1 Report

Overall Recommendation: Reject (article has serious flaws, additional experiments needed, research not conducted correctly)

This study investigated the use of antimicrobials in food producing animals from 2018 to 2020 in Nepal. This study is useful for risk analysis, planning, interpreting resistance surveillance data, and evaluating the effectiveness of prudent use efforts and mitigation strategies. However, this study investigated only the usage of antimicrobials; antimicrobial resistance patterns and/or resistance genes were not evaluated. In addition, the investigation's specifics are vague.

1. The author categorized administration routes of antimicrobials as injection/oral however, the oral routes should be classified as feed / water intake.

2. In addition, the type and amount of antibiotics used by animals varies, but no research was undertaken for this article on the sales of antimicrobials by animals.

3. For evaluating the effectiveness of prudent use efforts and mitigation strategies, period of the study should be longer.

4. To interpret resistance surveillance data, antimicrobial resistance phenotypes and genotypes should be investigated.

5. There was no data for the purpose of antimicrobials used in Nepal during period of the study.

The results of this article fit with the antimicrobial usage survey report, not with the paper itself, and are considered inappropriate for publication in this journal.

Reviewer 2 Report

Overall, the paper provides valuable insights into the quantities and types of antibiotics available in Nepal for food-producing animals, as well as the trends in antibiotic usage over the past three years. The study establishes a benchmark for future monitoring of antimicrobial usage in food-producing animals in Nepal and is useful for risk analysis, planning, interpreting resistance surveillance data, and evaluating the effectiveness of prudent use efforts and mitigation strategies.

However, there are some areas where the paper can be improved:

1The introduction provides a good background on the importance of the livestock industry in Nepal and the role of antimicrobials in mitigating infectious diseases in livestock. However, it would be helpful to explicitly state the main aim and research question(s) of the study to provide readers with a clear understanding of the scope of the paper.

2The methodology section could benefit from a more detailed description of the survey methods used to collect data from the different stakeholders, including the types of questions asked and how the data were analyzed.

3The discussion section could be expanded to provide more detailed insights into the implications of the findings. For example, the paper could discuss the potential impact of the decreasing trend in the use of critically important antibiotics and the implications for antimicrobial resistance in Nepal.

4The paper could also benefit from a more thorough discussion of the limitations of the study, including potential biases in the data collection process, limitations of the data sources, and assumptions made in the analysis.

5Finally, the paper would benefit from a more detailed conclusion that summarizes the main findings of the study and highlights the implications for future research and policy development.

Round 2

Reviewer 1 Report

I understand the aim of study, and the limitation that Nepal do not have legislations that require end useres to record the antimicrobials used in animals.

The results of this article fit with the antimicrobial usage survey report, not with the paper itself, and are considered inappropriate for publication in this journal.